# Observing ground-state properties of the Fermi-Hubbard model using a scalable algorithm on a quantum computer

Stasja Stanisic[1], Jan Lukas Bosse[1,2], Filippo Maria Gambetta[1], Raul A. Santos[3], Wojciech Mruczkiewicz [4], Thomas E. O'Brien[4], Eric Ostby[4] & Ashley Montanaro [1,2] ✉

The famous, yet unsolved, Fermi-Hubbard model for strongly-correlated electronic systems is a prominent target for quantum computers. However, accurately representing the Fermi-Hubbard ground state for large instances may be beyond the reach of near-term quantum hardware. Here we show experimentally that an efficient, low-depth variational quantum algorithm with few parameters can reproduce important qualitative features of medium-size instances of the Fermi-Hubbard model. We address $1 \times 8$ and $2 \times 4$ instances on 16 qubits on a superconducting quantum processor, substantially larger than previous work based on less scalable compression techniques, and going beyond the family of 1D Fermi-Hubbard instances, which are solvable classically. Consistent with predictions for the ground state, we observe the onset of the metal-insulator transition and Friedel oscillations in 1D, and antiferromagnetic order in both 1D and 2D. We use a variety of error-mitigation techniques, including symmetries of the Fermi-Hubbard model and a recently developed technique tailored to simulating fermionic systems. We also introduce a new variational optimisation algorithm based on iterative Bayesian updates of a local surrogate model.

Understanding systems of many interacting electrons is a grand challenge of condensed-matter physics[1]. This challenge is motivated both by practical considerations, such as the design and characterisation of novel materials[2], and by fundamental science[3–5]. Yet classical methods are unable to represent the quantum correlations occurring in such systems efficiently, and accurately solving the many-electron problem for arbitrary large systems is beyond the capacity of the world's most powerful supercomputers.

This problem is thrown into sharp relief by the iconic Fermi-Hubbard model[6,7], the simplest system that includes non-trivial correlations not captured by classical methods (e.g. density functional theory). Although a highly simplified model of interacting electrons in a lattice, to date the largest Fermi-Hubbard system which has been solved exactly consisted of just 17 electrons on 22 sites[8]. Approximate methods can address much larger systems, but suffer from significant uncertainties in computing physically relevant quantities in certain regimes[1].

Quantum computers can represent quantum systems natively, and may enable the solution of physical problems that classical computers cannot handle. The Fermi-Hubbard model has been widely proposed as an early target for quantum simulation algorithms[9–16]. As well as its direct application to understanding technologically-relevant correlated materials, the regularity and relative simplicity of the Fermi-Hubbard Hamiltonian suggest that it may be easier to solve using a quantum computer than, for example, a large unstructured molecule; on the other hand, the challenge that it presents for classical methods makes it an excellent benchmark for quantum algorithms.

[1]Phasecraft Ltd., Bristol, UK. [2]School of Mathematics, University of Bristol, Bristol, UK. [3]Phasecraft Ltd., London, UK. [4]Google Quantum AI, Mountain View, CA, USA. ✉e-mail: ashley@phasecraft.io

Small-scale experiments have used quantum algorithms to find ground states of the interacting Fermi-Hubbard model for instances on up to 4 sites[17–19] using up to 4 qubits. These experiments compress the model based on its symmetries; methods of this form, while having running time scaling polynomially with system size, are complex enough that solving a post-classical Fermi-Hubbard instance would not be viable on a near-term quantum computer.

Here we instead use an extremely efficient quantum algorithm, proposed in Ref. [12] based on previous work[10,11,20], to study medium-scale instances of the Fermi-Hubbard model without the need for compression. The algorithm fits within the framework of the variational quantum eigensolver[21,22] (VQE) using the Hamiltonian variational ansatz[10]. Based on extensive classical numerics for Fermi-Hubbard instances on up to 12 sites[12], this algorithm may be able to find accurate representations of the ground state of Fermi-Hubbard instances beyond classical exact diagonalisation by optimising over quantum circuits where the number of ansatz layers scales like the number of sites, corresponding to several hundred layers of two-qubit gates. While substantially smaller than previous quantum circuit complexity estimates for post-classical simulation tasks, this is still beyond the capability of today's quantum computers.

In this work, we demonstrate that a far lower number of ansatz layers can nevertheless reproduce qualitative properties of the Fermi-Hubbard model on quantum hardware. We apply VQE to Fermi-Hubbard instances on $1 \times 8$ and $2 \times 4$ lattices, using a superconducting quantum processor[23], and observe physical properties expected for the ground state, such as the metal-insulator transition (MIT), Friedel oscillations, decay of correlations, and antiferromagnetic order. These results rely on an array of error-mitigation techniques that improve substantially the accuracy of estimating observables on noisy quantum devices, opening the path to useful applications in the near future.

## Results

### Variational algorithm

Our algorithms attempt to approximate the ground state of the Fermi-Hubbard model,

$$H = -\sum_{\langle i,j \rangle, \sigma} \left( a_{i\sigma}^\dagger a_{j\sigma} + a_{j\sigma}^\dagger a_{i\sigma} \right) + U \sum_i n_{i\uparrow} n_{i\downarrow}, \qquad (1)$$

where $a_{i\sigma}(a_{i\sigma}^\dagger)$ is a fermionic operator that destroys (creates) a particle at site $i$ with spin $\sigma$, $n_{i\sigma} = a_{i\sigma}^\dagger a_{i\sigma}$ is the number (density) operator, and $\langle i,j \rangle$ denotes adjacent sites on a rectangular lattice.

Representing the Fermi-Hubbard Hamiltonian on a quantum computer requires a fermionic encoding. Here we use the well-known Jordan-Wigner transform, under which each fermionic mode maps to one qubit, interpreted as lying on a 1D line. This parsimony in space comes at the price that, except in 1D, some terms correspond to operators acting on more than two qubits:

$$a_i^\dagger a_j + a_j^\dagger a_i \mapsto \frac{1}{2}(X_i X_j + Y_i Y_j) Z_{i+1} \cdots Z_{j-1}, \qquad (2)$$

$$n_i n_j = a_i^\dagger a_i a_j^\dagger a_j \mapsto |11\rangle \langle 11|_{ij}. \qquad (3)$$

For $L_x \times L_y$ instances with $L_x \geq 2$, the "snake" ordering shown in Fig. 1a (for $2 \times 4$) can be used to map the rectangular lattice to a line. Under this mapping, horizontal terms only involve pairs of qubits, but some vertical terms act on larger numbers of qubits. As onsite terms always only involve pairs of qubits, we can place the qubits corresponding to spin-down modes after those corresponding to spin-up without incurring any additional cost for these long-range interactions.

The variational approach we use optimises over quantum circuits of the following form[12] (Fig. 1d). First, prepare the ground state of the noninteracting ($U = 0$) Fermi-Hubbard model, which can be achieved efficiently via a sequence of Givens rotations[11], which act on pairs of adjacent modes. Then repeat a number of layers, each consisting of time-evolution according to terms in the Fermi-Hubbard model.

The Hamiltonian $H$ has a natural decomposition into at most 5 sets of terms on a rectangular lattice such that all the terms in each set act on disjoint modes. This, in principle, allows the corresponding time-evolution steps to be implemented in parallel, although care must be taken over overlapping $Z$-strings in the Jordan-Wigner transform. Evolution times are variational parameters which are optimised using a classical optimisation algorithm. Within each layer, the terms within each set evolve for the same amount of time. For a $1 \times L_y$ instance, $L_y \geq 3$, each layer then has 3 parameters (one onsite term, and two types of hopping terms); for a $2 \times L_y$ instance, $L_y \geq 3$, each layer has 4 parameters; and for a $L_x \times L_y$ instance, $L_x, L_y \geq 3$, each layer has 5 parameters.

This structure is advantageous in two respects: the small number of parameters reduces the complexity of the variational optimisation process, and the variational ansatz respects the symmetries of the Fermi-Hubbard model, which (as we will see) provides opportunities for error mitigation. The same decomposition of $H$ into at most 5 parts allows for highly efficient measurement of energies using only 5 distinct measurements, each implemented via a computational basis measurement with at most one additional layer of two-qubit gates[12].

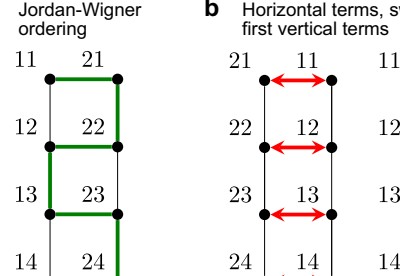

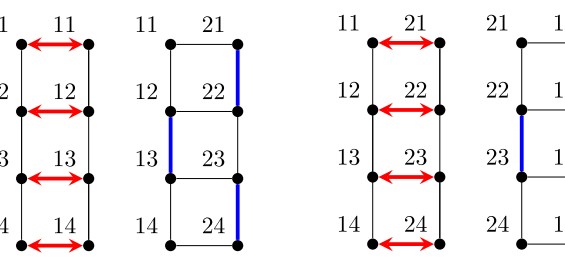

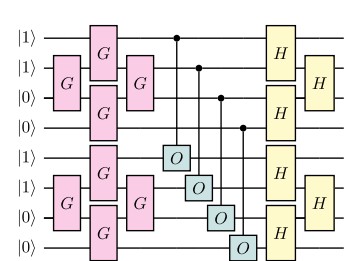

**Fig. 1 | Implementation of the Efficient Hamiltonian Variational ansatz.**
**a** Jordan-Wigner encoding mapping one spin sector of a $2 \times 4$ lattice to a line. Mapping is repeated for the other spin sector. **b**, **c** Horizontal terms are implemented combined with fermionic swaps (red); then the first set of vertical terms (blue); then another layer of fermionic swaps; then the second set of vertical terms. **d** Quantum circuit structure shown for a $1 \times 4$ instance at half-filling with one variational layer (actual experiments used up to 16 qubits). $G$: Givens rotations; $O$: onsite gates; $H$: hopping gates. Onsite and hopping gates correspond to time-evolution according to onsite and hopping terms in the Fermi-Hubbard

Hamiltonian; the structure of this part is repeated for multiple layers. All onsite terms have the same time parameter, and for $1 \times L_y$ instances, all hopping terms occurring in parallel have the same time parameter. When implemented on hardware in a zig-zag configuration, a layer of FSWAP gates is required before and after the onsite gates. First four qubits represent spin-up modes, last four represent spin-down modes. All operations in this diagram are implemented using two hardware-native two-qubit gates. Circuit is repeated multiple times for energy measurement, with differing measurement transformations at the end.

The final component of the VQE framework is the classical optimisation routine that optimises over the parameters in the quantum circuit to attempt to minimise the energy, and hence produce the ground state. This optimisation process is challenging as measurements are noisy, due to statistical noise and to errors in the quantum hardware. Here we introduce a new algorithm for this optimisation procedure, which we call BayesMGD. It enhances the MGD (Model Gradient Descent) algorithm[24,25] by performing iterative, Bayesian updates of a quadratic, local surrogate model to the objective function to make optimal use of the information gained from noisy measurements at each time step of the algorithm. During each iteration, the prior knowledge of the local quadratic fit to the objective function is updated by evaluating the latter in a neighbourhood of the current parameters. The gradient of this improved quadratic fit is then used to perform a gradient descent step. This is different from ordinary Bayesian optimisation, as used in[26–28], which uses a non-parametric global surrogate model and global acquisition functions to find the next evaluation point instead of gradient descent and the local surrogate models that we use. See section 4 of Supplementary Note 8 for details of experimental results comparing BayesMGD, MGD and SPSA (Simultaneous Perturbation Stochastic Approximation[29]).

## Quantum circuit implementation

We carried out our experiments on the "Rainbow" superconducting quantum processor in Google Quantum AI's Sycamore architecture, which had 23 qubits available in the configuration shown in Fig. 2.

We studied Fermi-Hubbard model instances on lattices of shapes $1 \times L_y$ and $2 \times L_y$. A $1 \times L_y$ Fermi-Hubbard system can be mapped to a $2 \times L_y$ rectangular qubit lattice by associating each site with two adjacent qubits for spin-up and spin-down. All hopping and onsite interactions can be implemented locally, leading to a very efficient quantum circuit. However, on the hardware platform we used, this configuration would only support a lattice of size at most $1 \times 4$. To enable us to study systems of size up to $1 \times 8$, we used a "zig-zag" configuration consisting of two nearby lines of length 8 (Fig. 2). Hopping interactions are implemented as local operations within each line, but onsite interactions are non-local and require a layer of swap operations.

For a $2 \times L_y$ lattice, due to the Z-strings occurring in the Jordan-Wigner transform, implementing some of the vertical hopping interactions directly would require 4-qubit operations. To remove the need for these, we use a fermionic swap (FSWAP) network[20]. A FSWAP operation rearranges the Jordan-Wigner ordering such that operations that were previously long-distance can now be implemented via two-qubit gates. Here, swapping across the horizontal direction of the lattice allows vertical interactions to be implemented efficiently (Fig. 1). The overhead for a $2 \times L_y$ lattice is only one additional layer of FSWAP gates per layer of the variational ansatz, together with some additional FSWAPs for measurement. However, using the FSWAP network approach does restrict the order in which terms are implemented, as vertical interactions occur across pairs determined by the Jordan-Wigner ordering. We therefore give this variational ansatz a specific name, the Efficient Hamiltonian Variational (EHV) ansatz[12].

In terms of quantum circuit complexity, the most complex instances we address are at or near half-filling, where with one variational layer, a $1 \times 8$ instance requires two-qubit gate depth at most 26 and at most 140 two-qubit gates, and a $2 \times 4$ instance requires two-qubit gate depth at most 32 and at most 176 two-qubit gates. For further implementation details, see Methods section.

## Error mitigation

Achieving accurate results requires a variety of error-mitigation procedures, divided into three categories. First, we use low-level circuit optimisations tailored to the hardware platform. Second, we take advantage of the symmetries of the Fermi-Hubbard Hamiltonian. Finally we use a technique for mitigating errors in fermionic Hamiltonian simulation algorithms. We explain these below.

We begin by optimising the quantum circuit to contain alternating layers of one-qubit and two-qubit gates, and selecting a high-performance set of qubits to use based on an initial test. We then use a technique based on spin-echo[30] where every other layer of two-qubit gates is sandwiched between layers of X gates on every qubit. This led to a substantial reduction in error in our experiments, which we attribute to two possible causes: that these X gates are inverting single-qubit phase errors that accumulate during the circuit; and that they modify "parasitic CPHASE" errors occurring on the two-qubit gates, which are known to be substantial[31].

The symmetry-based techniques for error mitigation that we use exploit number conservation per spin sector, time reversal, particle-hole and lattice symmetries. Number conservation allows error-detection by discarding runs where final and initial occupations do not match. In particular, this detects many errors that occur due to incorrect qubit readout, a significant source of error in superconducting qubit systems. In our 16-qubit experiments, we observed that between 7% and 29% of runs were retained; see Supplementary Note 4 for further discussion of sampling overhead and remarks on scalability, for this and other techniques. The other three symmetries allow us to average results obtained from a state and its symmetry-transformed partner.

The last error-mitigation technique we used is targeted at quantum algorithms for general fermionic systems[32], and is called Training with Fermionic Linear Optics (TFLO). TFLO uses efficient classical simulation of quantum circuits of time-evolution operations by quadratic fermionic Hamiltonians[33] (so-called fermionic linear optics (FLO) circuits). Expectations of energies, or other observables of interest, for states produced by FLO circuits can be calculated exactly classically, and approximately using the quantum computer. These pairs of exact and approximate energies can be used as training data to infer a map from approximate energies computed by the quantum computer, at points which are not accessible classically, to exact

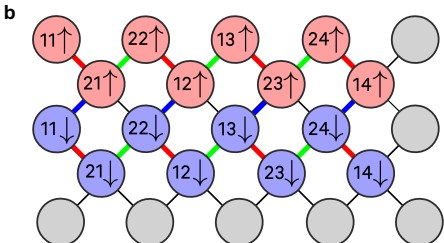

**Fig. 2 | Qubit layout for implementing two Fermi-Hubbard instances. (a)** $1 \times 8$ instance, **(b)** $2 \times 4$ instance. In each case two qubits are used to encode each site. Operations between qubits in variational layers occur in the following pattern. $1 \times 8$: blue (FSWAP), red (onsite), blue (FSWAP), red (vert1), green (vert2). $2 \times 4$: blue (FSWAP), red (onsite), blue (FSWAP), red (horiz + FSWAP), green (vert), red (FSWAP), green (vert). Vertical interaction parameters for $2 \times 4$ depend on the parity of the row index. Grey circles denote the unused qubits on the 23-qubit Rainbow chip.

energies. For this map to be accurate, the FLO circuits should approximate the real circuits occurring in the algorithm.

TFLO is ideally suited to the Fermi-Hubbard model, as most of the operations in the VQE circuit are FLO operations, including initial state preparation, time-evolution by the hopping terms, and measurement transformations. The only operations in the circuit that are not FLO are time-evolution by the onsite terms. Therefore, we can find a suitable training set by choosing arbitrary parameters for the hopping terms and setting the parameters of the onsite terms to 0. Compared with previous implementations[32], here we improve accuracy by choosing these parameters carefully to maximise their spread, using a linear fitting algorithm designed to handle outliers[34,35], and implementing a final step which aims to correct residual error. More details on all our error mitigation techniques are included in Supplementary Note 4, and results are shown in Fig. 3b. As expected, in some cases the inclusion of an additional error-mitigation technique can actually increase the level of error. One explanation for this is that our error-mitigation methods are based on using additional data at other parameter values (for example by averaging or linear interpolation), and hence if the result before mitigation happens to be accurate, error-mitigation could make it worse.

## Physical observables

We used the BayesMGD algorithm within the VQE framework to determine the parameters required to produce approximate ground states of instances of the Fermi-Hubbard model on up to 8 sites, by minimising the energy expectation value calculated from the state produced by the VQE circuit on the quantum processor. Once these parameters are determined, we have a quantum circuit to produce this state–which we call the VQE ground state below–and can perform measurements to determine its properties. We found that BayesMGD was able to converge on parameters corresponding to the VQE ground state within a small number of iterations (Fig. 3). Interestingly, BayesMGD was able to further improve the parameters, as measured by the exact, simulated energy at the parameters $\theta_k$, even when the energy estimates from samples increased instead of decreasing. This behaviour can be seen in Fig. 3a and its inset. We attribute this phenomenon to the performance of the device changing during the optimisation process. As a local, gradient-based optimiser which is constantly updating its parameters, BayesMGD is immune to certain global fluctuations of the optimisation landscape, for example shifting by an overall additive or multiplicative constant.

First we compute the energy in the VQE ground state for $1 \times 8$, $2 \times 4$ and $1 \times 4$ systems for all occupation numbers $1 \leq N_{occ} \leq 15$ ($\leq 7$ for the $1 \times 4$ system) (Fig. 4). In all cases good quantitative agreement is achieved with the exact lowest energy achievable with 1 layer of the VQE ansatz. To validate that the quantum algorithm goes beyond what is achievable with a straightforward classical ansatz, we compare with energies achieved by optimised Slater determinant states (see Supplementary Note 5). Further, in the $1 \times 4$ case (Fig. 4c), lower energy is achieved with a 2-layer variational ansatz than is theoretically possible with 1 layer, demonstrating that increased ansatz depth can lead to higher performance. In general the lowest energy achievable with 1 VQE layer is larger than the energy of the first excited state, so energies alone do not certify that we have prepared a good approximation of the true ground state. However, the VQE ground state achieves non-trivial fidelity with the true ground state, in theory, and usually larger fidelity than the best achievable with a Slater determinant; for example, fidelity ≈ 0.77 for $1 \times 8$ at half-filling. See Supplementary Note 5 for details of VQE and Slater determinant fidelities in other cases and section 1 of Supplementary Note 7 for a separate analysis of the hopping and onsite energies in the best possible VQE states.

Next, we study the onset of the MIT[36] between half-filling and away from half-filling in a $1 \times 8$ system (Fig. 4). Although in finite systems

there is no true phase transition, we concentrate on two signals that are a precursor to this transition. First, a Mott gap which increases with $U$, shown by a nonzero derivative of the chemical potential (i.e. the second derivative of the energy, here approximated as $E(N_{occ}+1) + E(N_{occ}-1) - 2E(N_{occ})$) at half-filling ($N_{occ} = 8$), when $U \neq 0$ (see insets in Fig. 4d, e). The physical origin of this can be understood as the energy penalty imposed for adding an electron on top of a half-filled state, where all sites are occupied. While in a 1D system of size $L_x$ the energy difference between states with occupations away from half-filling scales as $O(1/L_x)$, a fixed gap to charged excitations is a unique characteristic of a Mott insulator. Second, we observe the spatial decay of normalised charge correlations with distance from the first site, $C^c(1,i) := (\langle n_1 n_i \rangle - \langle n_1 \rangle \langle n_i \rangle)/(\langle n_1^2 \rangle - \langle n_1 \rangle^2)$ (Fig. 4f), where $n_i = n_{i\uparrow} + n_{i\downarrow}$. The steepest decay appears at half-filling ($N_{occ} = 8$), where the Mott gap implies the exponential decay of correlations. Further away from half-filling, the slower decay is a signature of increased conductivity. We have also computed these quantities for a $2 \times 4$ system, where the results are suggestive but the MIT is not clear (see section 2 of Supplementary Note 7).

Following, we study the behaviour of charge and spin densities at different sites and occupation numbers (Fig. 5). Boundaries in a finite-size system break the translational invariance and, as a consequence, induce Friedel oscillations in the charge density of the ground state[37] with twice the Fermi wavevector $k_F$. Therefore, in a 1D system with even (odd) occupation number $N_{occ}$, they result in a ground-state charge density profile with $N_{occ}/2$ ($(N_{occ}+1)/2$) peaks. Evidence of this behaviour can be clearly seen in the VQE results in Fig. 5(a). On the other hand, for strong onsite interactions and/or low fillings Wigner oscillations with wavevector $4k_F$ are also expected as a consequence of the Coulomb repulsion[38,39]. In 1D, the latter are responsible for the emergence of $N_{occ}$ peaks in the ground-state charge density and are indeed visible in Fig. 5e, especially for $N_{occ} \leq 4$. Hence, a comparison between Fig. 5a, e suggests that a higher-depth variational ansatz is required to fully capture strong interaction effects. We see that, following error mitigation, the density in the case of equal number of spin-up and spin-down electrons is indeed close to zero (Fig. 5b) as expected from symmetry, compared with the more substantial densities for odd occupations (Fig. 5c, d), which in our case always corresponds to including an extra spin-up particle. These densities display a similar structure to the charge densities at the corresponding occupation.

To explore the differences between 1D and 2D, we compute (Fig. 6) the spin correlations $C^s(i,j) := \langle S_i^z S_j^z \rangle - \langle S_i^z \rangle \langle S_j^z \rangle$, where $S_i^z = n_{i\uparrow} - n_{i\downarrow}$, in the VQE ground state at half-filling for $1 \times 8$ and $2 \times 4$ lattices with strong onsite interaction ($U = 4$). We observe antiferromagnetic correlations compatible with the expected behaviour for that size, which are stronger in 2D compared with 1D. Antiferromagnetic and charge-density-wave ordering around half-filling are expected features of the Mott state in 2D. The charge profile for the $2 \times 4$ system is reported in Supplementary Fig. 6 (see also discussion therein). We also explore the antiferromagnetic character of the ground state for different onsite interactions and for occupations 7 and 8 (Supplementary Fig. 5) where we observe that the system is indeed less antiferromagnetic at $N_{occ} = 7$ than at half-filling for $U = 4, 8$. Although the VQE ground state does not capture the value of the total staggered spin correlation in the true ground state quantitatively, it does follow the same trend.

## Discussion

We have shown that fundamental qualitative features of medium-size instances of the Fermi-Hubbard model, using a number of qubits 4 times larger than Fermi-Hubbard experiments previously reported in the literature[17–19], can be extracted using a quantum computer with a low-depth variational ansatz and techniques that mitigate the inherent noise of near-term quantum hardware. To do this, we implemented a quantum circuit based on the structure of the Hamiltonian that

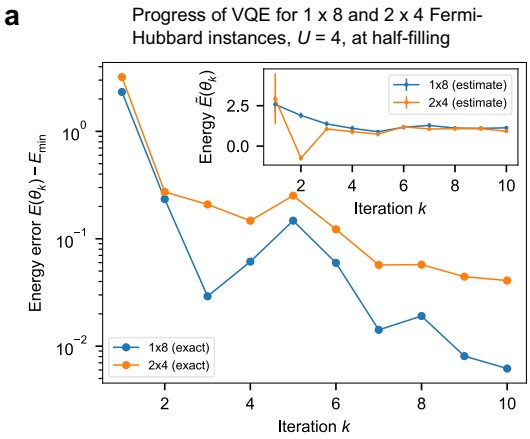

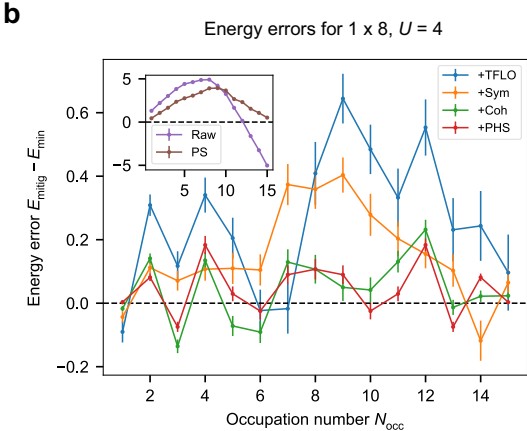

**Fig. 3 | Experimental results for the BayesMGD algorithm and final energy errors with respect to the VQE ground state. a** Progress of VQE for $1 \times 8$ and $2 \times 4$ Fermi-Hubbard instances at half-filling, as measured by the error between energy at parameters $\boldsymbol{\theta}_k$ and VQE ground energy $E_{min}$ (main plot log scale, inset linear scale). "Estimate" is the energy estimated by the BayesMGD algorithm during the VQE procedure based on measurement results, "exact" is the true energy at the corresponding parameters. **b** Final errors in measured energy following error mitigation on the final state. "Raw": no error mitigation. "PS": only postselection on occupation

number. "+Sym": also time-reversal symmetry. "+TFLO": also Training with Fermionic Linear Optics[32]. "+Coh": also coherent error correction in TFLO. "+PHS": also particle-hole symmetry. Raw/PS shown with different scale for clarity. Reduction in error using all techniques is e.g. ~46× at half-filling. Each error mitigation method is applied as well as all previous methods. Plots show a piecewise linear interpolation between integer occupations. Error bars were calculated according to the procedure described in Methods section and are often too small to be visible.

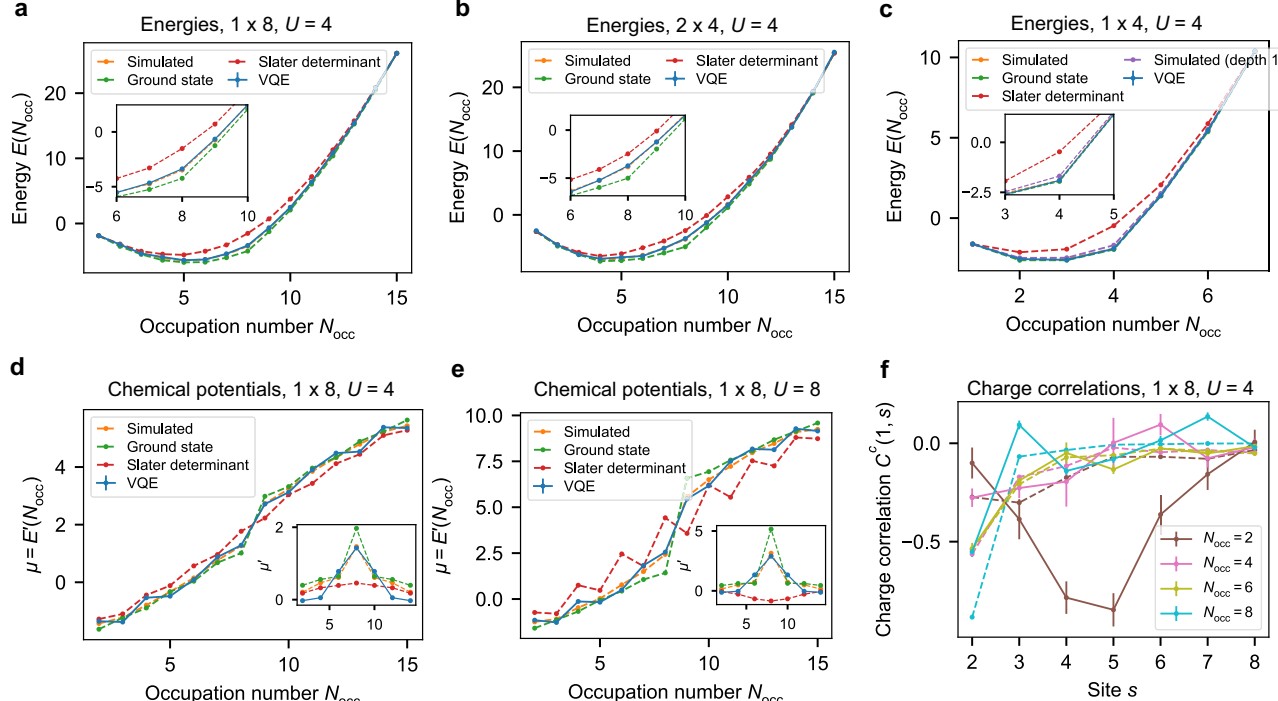

**Fig. 4 | Experimental energies, chemical potentials and charge correlations.** "VQE": experimental data. "Simulated": the lowest energy achievable in the VQE ansatz. "Ground state": energy in the true ground state within each occupation number subspace. "Slater determinant": the energy achieved by an optimised Slater determinant state as detailed in Supplementary Note 5. Dashed lines are exact numerical calculations, solid line is experimental data. Plots show a piecewise linear interpolation between integer occupations. **a**–**c** Energies $E(N_{occ})$ produced by VQE experiments compared with exact results ($U = 4$). VQE results for $1 \times 8$ and $2 \times 4$ use one variational layer; $1 \times 4$ has two variational layers. Inset shows zoomed-in region around half-filling. **d, e** Chemical potentials $\mu$ for a $1 \times 8$ system, where $\mu(N_{occ}) = E(N_{occ}) - E(N_{occ} - 1)$. Inset shows the derivative $\mu'(N_{occ}) = E(N_{occ} + 1) - 2E(N_{occ}) + E(N_{occ} - 1)$ of the chemical potential at even occupations. **f** Decay of normalised charge correlations $C^c(1, i) = (\langle n_1 n_i \rangle - \langle n_1 \rangle \langle n_i \rangle)/(\langle n_1^2 \rangle - \langle n_1 \rangle^2)$ for even occupation numbers. Solid lines: experimental results. Dashed lines: correlations in ground state. Error bars were calculated according to the procedure described in Methods section and are often too small to be visible.

incorporates the restrictions due to the fermionic Jordan-Wigner string in an efficient way, and deployed a range of error mitigation techniques to extract a meaningful signal from the noisy measurements on the quantum device. This allowed us to compute energies relatively accurately for states that can be produced with one variational layer. It is interesting to note that the features we observe are visible despite the fidelity between the VQE ground state and the true ground state—which is a very stringent measure of closeness—being low enough that, in principle, these features might not still be present. It is also worth noting that the error in the measured energy

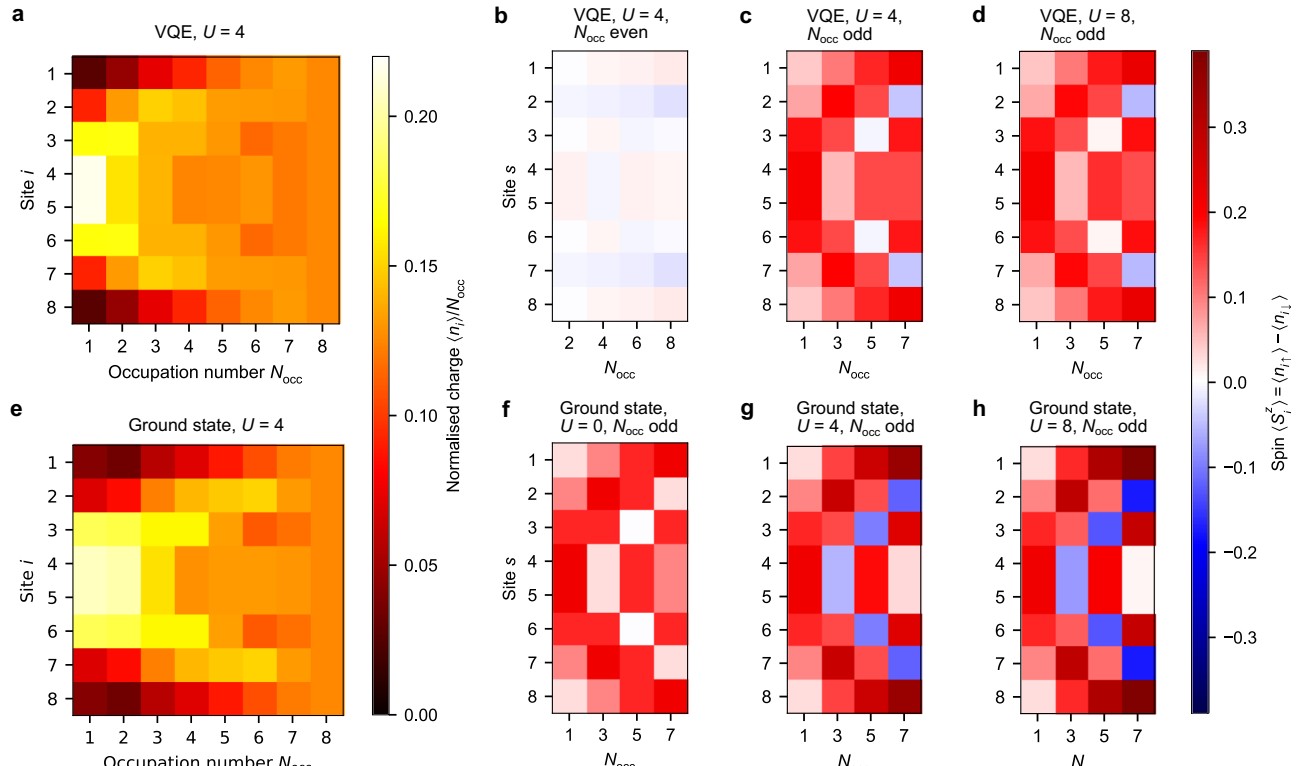

**Fig. 5 | Charge and spin densities for a 1 × 8 lattice.** In all panels the $X$ axis gives the occupation number $N_{occ}$ while the $Y$ axis gives the site index. The top row shows experimental VQE results, while the bottom one displays exact ground state quantities. **a**, **e** charge density. **b–d**, **f–h** Spin density. Here, plots are split by even/odd occupations. In the ground state, spin is 0 everywhere for $N_{occ}$ even.

per site compared with the ideal VQE state energy per site at depth one that we find is not too far from the discrepancy encountered between state of the art methods, although those can reach larger sizes (see Supplementary Note 9 for a comparison). This hints that being able to scale up the system size and the number of layers in VQE, while preserving the same level of errors, could make it competitive with state of the art classical approaches.

We expect that the use of a higher-depth variational ansatz in larger systems will enable the demonstration of phenomena such as Wigner oscillations, charge-density-wave ordering, and magnetic instabilities, and will shed some light on the different phases of the 2D system. Achieving a high level of quantitative accuracy in computing true ground state energies is a more significant challenge, which we expect will require a larger number of variational layers still, perhaps scaling with the system size[12]. Our efficient algorithm and error-mitigation techniques provide a template that can readily be scaled up to larger systems as quantum computing hardware continues to improve.

## Methods

### Implementation of the Efficient Hamiltonian Variational ansatz

The variational ansatz we used is based on the Hamiltonian Variational ansatz[10], but with some of the hopping terms implemented using FSWAP networks. This leads to the terms being implemented in a particular, fixed order, which can affect the performance of the quantum algorithm[12]. We therefore refer to this ansatz specifically as the EHV ansatz.

There are five operations that we need as building blocks for our circuit, each of which is implemented using two hardware-native $\sqrt{\text{iSWAP}}$ gates and some single-qubit gates. The initial state is prepared using Givens rotations (gate G in Fig. 7). Then each layer of the EHV ansatz consists of onsite (gate O in Fig. 7) and hopping (gate H in Fig. 7) gates, corresponding to time-evolution by onsite and hopping terms in

the Fermi-Hubbard Hamiltonian of Eq. (1), respectively, where hopping terms are assumed to act only on adjacent modes in the Jordan-Wigner transform. We have

$$\text{H}(\theta) = e^{-i\theta(XX+YY)/2}, \quad \text{O}(\phi) = e^{i\phi|11\rangle\langle11|}.$$

For a lattice with shape $2 \times L_y$, we need a fermionic SWAP (FSWAP) gate to implement the FSWAP network (gate FSWAP in Fig. 7). Finally, we need a gate for the change of basis needed to measure the hopping terms (gate B in Fig. 7). In previous work it was suggested to use a Hadamard gate within the $\{|01\rangle, |10\rangle\}$ subspace[12]; here we use an equivalent operation that can be implemented more easily. Note that this operation preserves occupation number, which allows the use of error detection.

When implemented on hardware, the single-qubit gates shown in Fig. 7 are decomposed in terms of the hardware-native PhasedXZ gate primitive. Due to a sign error in our implementation of this decomposition, in the experiments the onsite gate O($\phi$) was implemented up to identical single-qubit Z rotations on each qubit, which leave the overall state unchanged within a fixed occupation number subspace. Spot checks comparing with a correctly decomposed onsite gate confirmed that, as expected, these Z rotations did not affect the overall accuracy of the experiment.

The first step of the EHV ansatz is to prepare the ground state of the noninteracting ($U = 0$) Fermi-Hubbard Hamiltonian. Preparation of this state has been studied extensively before and an efficient algorithm using Givens rotations is known[11] which achieves circuit depth $N - 1$, and a total of $(N - N_\sigma)N_\sigma$ Givens rotations (for each spin sector), where $N$ is the number of modes per spin sector, or equivalently the size $L$ of the lattice ($L = L_x \times L_y$) and $N_\sigma$ is the number of fermions in the spin sector $\sigma$. A detailed analysis of alternative state preparation methods[12] concluded that this algorithm was the most efficient known

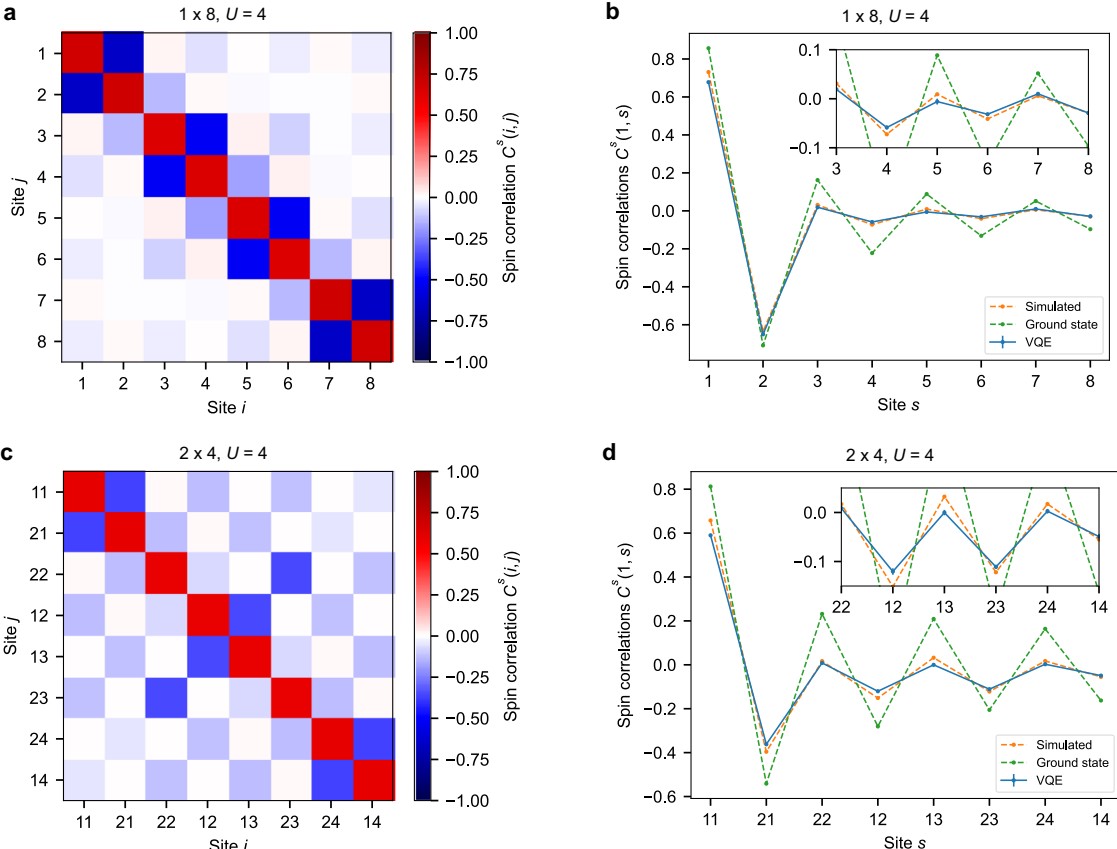

**Fig. 6 | Antiferromagnetic correlations at half-filling ($U = 4$) obtained with the quantum processor. a**, **c** Spin correlation function $C^s(i,j)$ for a $1 \times 8$ (**a**) and $2 \times 4$ (**c**) instance of the Fermi-Hubbard model. The ordering of sites for the $2 \times 4$ lattice follows the Jordan-Wigner "snake" (Fig. 1). **b**, **d** Spin correlation function $C^s(1, s)$ for a $1 \times 8$ (**b**) and $2 \times 4$ (**d**) system. The meaning of labels in **b**, **d** is as in Fig. 4.

for small system sizes. To prepare this initial state we use the OpenFermion[40] implementation of this algorithm.

In order to implement our algorithm, two types of swap operations are needed: FSWAPs to rearrange the Jordan-Wigner ordering, and physical (standard) swaps to bring distant qubits together. An FSWAP operation can be implemented with two native gates, as shown in Fig. 7, whereas physical swaps would require three native gates. However, in our experiment we are always able to use FSWAPs in place of physical swaps. The one place where physical swaps would naturally be used is rearranging qubits before and after implementing an onsite (CPHASE) gate. As the onsite gates are diagonal, the sign part of the FSWAP gates commutes with them and cancels out.

Measuring the energy of the VQE state can be achieved with three different measurement circuits for $1 \times L_y$ instances (vertical hopping 1 and 2, and onsite), and with four circuits for $2 \times L_y$ instances (horizontal hopping, vertical hopping 1 and 2, and onsite). Onsite energy is measured via a computational basis measurement and counting the number of sites where both spin-up and spin-down qubits receive a 1 outcome. For $1 \times L_y$, each type of vertical hopping term is measured using a layer of basis transformations, using the $B$ gate shown in Fig. 7. These gates diagonalise the hopping terms, enabling the corresponding energy to be measured via a computational basis measurement. The second type of vertical hopping measurement can be merged into the final layer of gates in the circuit (Fig. 1) to reduce the quantum circuit depth. Measuring the energy for $2 \times L_y$ instances is similar, except that vertical hopping terms are split up in a different way (also see Fig. 1), and some of them require a layer of FSWAP gates before measurement.

Additional details about circuit complexity and scaling of algorithms are given in Supplementary Note 3.

## Variational optimiser
In this work we introduce a new variational optimisation method, which we call Bayesian model gradient descent (BayesMGD), and compare it with the standard simultaneous perturbation stochastic approximation (SPSA) algorithm[29], which has been previously successfully used as an optimisation algorithm for VQE on superconducting quantum computers[41,42], and the model gradient descent (MGD) algorithm introduced by Sung et al.[24] for precisely the task of optimising parametric quantum circuits[43].

The main idea of MGD is to sample points and function values $(\boldsymbol{\theta}_i, y_i)$ in a trust region around $\boldsymbol{\theta}$, fit a quadratic surrogate model using linear least squares to all data available in the trust region and use this surrogate model to estimate the gradient. Our algorithm is designed to improve on these ideas via Bayesian analysis. We perform iterative, Bayesian updates on the surrogate model and utilise the sample variance to estimate the uncertainty in the fit parameters and surrogate model evaluations. Utilising the sample variance to estimate the uncertainty of function evaluations allows for more accurate surrogate models and estimating the uncertainty in the surrogate model evaluations allows us to put error bars on the predictions.

We are given a random field $f(\boldsymbol{\theta})$ (that is, a collection of random variables parameterised by $\boldsymbol{\theta}$) and want to find the parameters $\boldsymbol{\theta}$ such that the expectation value $\mu(\boldsymbol{\theta}) := \mathbb{E}[f(\boldsymbol{\theta})]$ is minimal. We assume that at each $\boldsymbol{\theta}$ the variance of the random variable $f(\boldsymbol{\theta})$ is finite, such that the central limit theorem is applicable to sample means of $f(\boldsymbol{\theta})$. Since we are always interested in situations where we take many samples at a given $\boldsymbol{\theta}$ and approximate $\mu(\boldsymbol{\theta})$ by their mean, we can equivalently assume—and will from now on—that $f(\boldsymbol{\theta})$ is normally distributed at each $\boldsymbol{\theta}$ with known variance $\sigma^2(\boldsymbol{\theta})$. Furthermore, the mean function $\mu(\boldsymbol{\theta})$ is assumed to be smooth and hence it is locally always well described by a

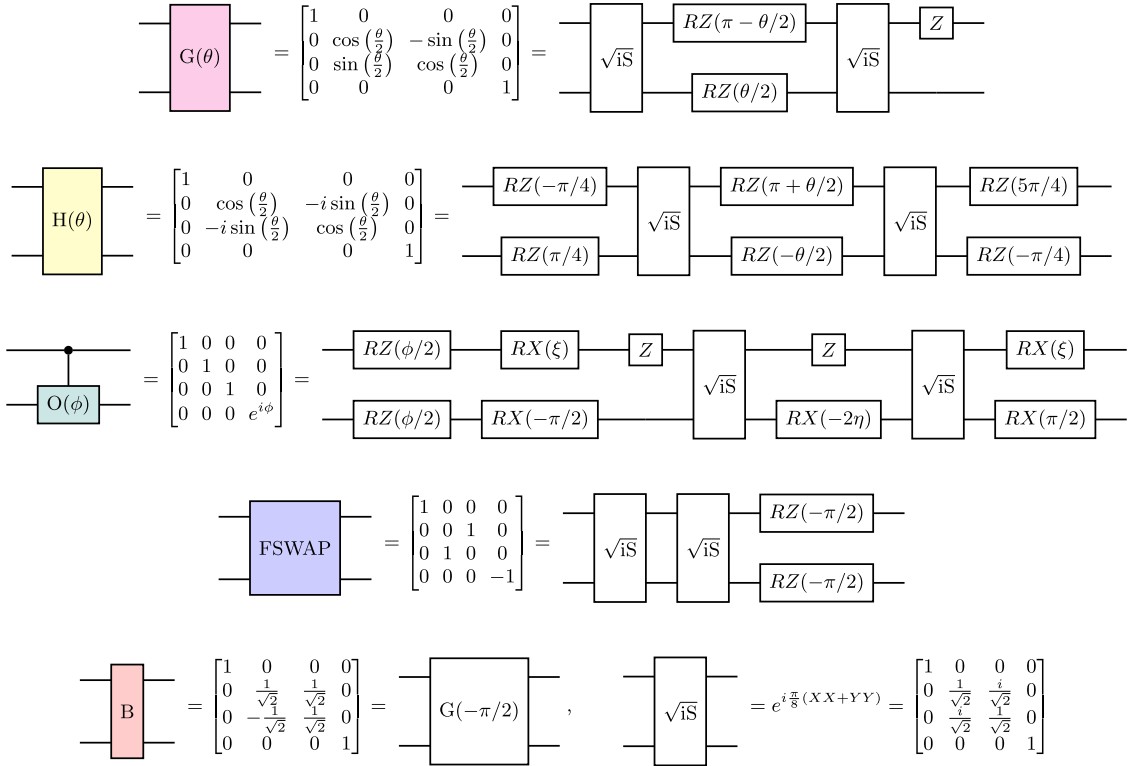

**Fig. 7 | Operations used within the Fermi-Hubbard VQE circuit.** From top to bottom, Givens rotations, hopping terms, onsite terms, fermionic swaps, and basis changes for hopping term measurement—and how they can be decomposed in terms of 1 and 2-qubit gates. Here, $\eta = \arcsin(\sqrt{2}\sin(\phi/4))$, $\xi = \arctan(\tan(\eta)/\sqrt{2})$, and $\phi \in [-\pi, \pi]$.

quadratic surrogate model

$$f_s(\boldsymbol{\theta}; \boldsymbol{\beta}) = \beta_0 + \sum_{j=1}^{n_c} \beta_j \theta_j + \sum_{j,k=1, j<k}^{n_c} \beta_{jk} \theta_j \theta_k \qquad (4)$$

which is linear in its model parameters $\beta_0$, $\beta_j$ and $\beta_{jk}$, and where $n_c$ is the number of circuit parameters.

In each iteration $m$ we randomly pick $n_p = \eta \dim(\boldsymbol{\beta})$ sampling points $\boldsymbol{\theta}^{(i)}$ in a $\delta_m$-ball around $\boldsymbol{\theta}_m$ and get noisy function evaluations $y_i \sim \mathcal{N}(\mu(\boldsymbol{\theta}^{(i)}), \sigma_i^2)$ with approximately known uncertainty $\sigma_i$, where $\eta$ is the ratio between the number of new sampling points $n_p$ and the number of points needed for a fully determined quadratic fit. The sampling radius scales as $\delta_m = \delta/m^\xi$ with a sample radius decay exponent $\xi$ and initial sampling radius $\delta$. This new data $\{\boldsymbol{\theta}^{(i)}\}$ and $\{y_i, \sigma_i\}$ is used to update our belief $p_{m|m-1}(\boldsymbol{\beta})$ about the parameters $\boldsymbol{\beta}$ at the $m$-th step given the data up until step $m-1$ using Bayes' rule to a new belief $p_{m|m}(\boldsymbol{\beta})$ incorporating the new data from the $m$-th step:

$$
\begin{aligned}
p_{m|m}(\boldsymbol{\beta}) &= P(\boldsymbol{\beta} \mid \{\boldsymbol{\theta}^{(i)}\}, \{y_i, \sigma_i\}) \\
&\propto P(\{y_i, \sigma_i\} \mid \{\boldsymbol{\theta}^{(i)}\}, \boldsymbol{\beta}) \, p_{m|m-1}(\boldsymbol{\beta}) \\
&= \prod_{i=1}^{n_p} \mathcal{N}(y_i; f_s(\boldsymbol{\theta}^{(i)}; \boldsymbol{\beta}), \sigma_i) \\
&\quad \times \mathcal{N}(\boldsymbol{\beta}; \boldsymbol{\beta}_{m|m-1}, \boldsymbol{\Sigma}_{m|m-1}) \\
&=: \mathcal{N}(\boldsymbol{\beta}; \boldsymbol{\beta}_{m|m}, \boldsymbol{\Sigma}_{m|m}),
\end{aligned}
\qquad (5)
$$

where in the last line we use the fact that the product of Gaussians is again a Gaussian to implicitly define $\boldsymbol{\beta}_{m|m}$ and $\boldsymbol{\Sigma}_{m|m}$. We defer the detailed derivation of $\boldsymbol{\beta}_{m|m}$, $\boldsymbol{\Sigma}_{m|m}$ in terms of the prior $\boldsymbol{\beta}_{m|m-1}$, $\boldsymbol{\Sigma}_{m|m-1}$ and new data to Supplementary Note 8, together with a discussion of the relation of BayesMGD and Kalman filters and pseudo-code for the algorithm.

Since the surrogate model $f_s(\boldsymbol{\theta}; \boldsymbol{\beta})$ is linear in the model parameters $\boldsymbol{\beta}$ the usual uncertainty propagation formulas are exact and we know that

$$f_s(\boldsymbol{\theta}_m; \boldsymbol{\beta}) \sim \mathcal{N}\left(f_s(\boldsymbol{\theta}_m; \boldsymbol{\beta}_{m|m}), (\nabla_{\boldsymbol{\beta}} f_s)^\dagger \boldsymbol{\Sigma}_{m|m} \nabla_{\boldsymbol{\beta}} f_s\right), \qquad (6)$$

where $\nabla_{\boldsymbol{\beta}} f_s$ denotes the gradient of $f_s$ with respect to $\boldsymbol{\beta}$ evaluated at $(\boldsymbol{\theta}_m; \boldsymbol{\beta}_{m|m})$. Similarly, we also obtain a distribution over the gradient $\nabla_{\boldsymbol{\theta}} f_s(\boldsymbol{\theta}_m; \boldsymbol{\beta})$. The maximum a posteriori estimate for the gradient is simply obtained by plugging the most likely value $\boldsymbol{\beta}_m$ for the model parameters $\boldsymbol{\beta}$ into the gradient of the surrogate model:

$$g(\boldsymbol{\theta}_m) = \nabla_{\boldsymbol{\theta}} f_s(\boldsymbol{\theta}; \boldsymbol{\beta}_{m|m}). \qquad (7)$$

With this estimate of the gradient we perform a gradient descent step

$$\boldsymbol{\theta}_{m+1} = \boldsymbol{\theta}_m - \gamma_m g(\boldsymbol{\theta}_m). \qquad (8)$$

Here, $\gamma_m = \gamma/(m+A)^\alpha$ is the gradient step width with a stability constant $A$, decay exponent $\alpha$ and initial step width $\gamma$.

Changing $\boldsymbol{\theta}$ does not change the local surrogate model, but it adds uncertainty proportional to the step width to it. Hence the belief at $\boldsymbol{\theta}_{m+1}$ without data at that point is described by

$$
\begin{aligned}
\boldsymbol{\beta}_{m+1|m} &= \boldsymbol{\beta}_{m|m} \\
\boldsymbol{\Sigma}_{m+1|m} &= \boldsymbol{\Sigma}_{m|m} + \frac{\gamma_m^2 |g(\boldsymbol{\theta}_m)|^2}{l^2} \mathbb{1},
\end{aligned}
\qquad (9)
$$

where $l$ is the length scale on which our quadratic model becomes invalid. The choice of adding uncertainty proportional to the squared step width is heuristic so far, but can be motivated using Gaussian processes. A Gaussian process is a probability distribution over

**Table 1 | BayesMGD characteristics in experiments**

| Instance | Params | Points/iter | Max evals | Max iters |
|---|---|---|---|---|
| 1 × 4 | 6 | 42 | 2520 | 30 |
| 1 × 8 | 3 | 15 | 300 | 10 |
| 2 × 4 | 4 | 23 | 600 | 14 |

1 × 4 instances had two variational layers, others had one variational layer.

functions that allows one, among other things, to compute the probability distribution of function value, gradient and hessian at some point $\theta_{m+1}$ conditioned on the function value, gradient, and hessian available at some previous point $\theta_m$. For a Gaussian process with a squared exponential kernel and small $|\theta_m - \theta_{m+1}|$ the uncertainty about function value, gradient, and hessian at $\theta_{m+1}$ grows with the squared distance from $\theta_m$. The exact rate at which the uncertainty for each of the entries of $\beta_m$ grows requires in-depth analysis that we replaced with uniform scaling in all entries.

The key novel ingredient in this algorithm is the usage of Bayesian methods to reuse data obtained in previous iterations of the optimisation and optimally incorporate the measurement uncertainty into the estimates of the cost function. If the ansatz circuits permit, the algorithm may be further improved by replacing the generic quadratic surrogate model with a more informed surrogate model that is motivated by the true analytical form of the cost function, e.g. the trigonometric polynomials used in the quantum analytic descent algorithm[44] when the ansatz gates are generated by tensor products of Pauli operators.

Note added—In independent recent work[45] another optimisation algorithm, called SGSLBO (Stochastic Gradient Line Bayesian Optimisation), is proposed for VQE that is at first glance similar to ours. However, this algorithm is based on the use of stochastic gradient descent to determine the gradient direction paired with Bayesian optimisation for a line search along the gradient direction. In our case "Bayesian" refers to the iterative Bayesian procedure we use to update the model parameters $\beta$.

### Details of implementation parameters

Characterising the VQE ground state for a given Fermi-Hubbard instance can be separated into two parts: the VQE part, which runs the BayesMGD algorithm to determine the optimal variational parameters for the quantum circuit; and the state preparation part, which uses these parameters to produce copies of the VQE ground state itself, and also many other FLO states used for error mitigation (see section 3 of Supplementary Note 4). These parts can be carried out at different times, which may be advantageous, as device performance fluctuates over time. The state preparation part uses all error mitigation techniques described in Supplementary Note 4, whereas for efficiency the VQE part does not use particle-hole symmetry or TFLO.

In all cases of the VQE part, the BayesMGD optimiser used 1000 shots (energy measurements) per evaluation point, multiplied by 2 for evaluating at the parameters and their negations (see section 2 of Supplementary Note 4). Bounds on numbers of evaluations are shown in Table 1. For all instances, hyperparameters $\eta = 1.5$, $\delta = 0.6$, $\xi = 0.101$, $l = 0.2$ were used. For 1 × 4 and 1 × 8, $\gamma = 0.3$, $A = 1$ were used, whereas for 2 × 4, $\gamma = 0.6$, $A = 2$ were used. Increasing $\gamma$ moves through the parameter space more aggressively, and increasing the stability parameter $A$ reduces the chance of overaggressive moves at the start of the algorithm. Wall clock time for completing a VQE run was under 30 min for 1 × 8 instances, and under 70 min for 2 × 4 instances. We split the circuits evaluated into batches of size at most 80 to avoid timeout and circuit size constraints imposed by the quantum cloud platform.

In the state preparation part, we compute the energy of the VQE ground state by taking the average over 100,000 energy measurements, again both at the VQE parameter values and their negations. In

order to use TFLO, we also evaluate the energy at the closest FLO point (the one where the onsite parameters are set to 0), with 100,000 energy measurements; and also at 16 other points (and their negations), which have been chosen such that their exact energies are well-spaced. For each of these 16 points we perform 20,000 energy measurements. We carried out this procedure three times for each instance. Wall clock times are up to ~8 min per run for 1 × 8 and 2 × 4 instances.

### Error analysis

Error bars for energies and other quantities computed using VQE were derived as follows. First, we assume that measurements of each observable—conditioned on the occupation number in each spin sector being correct—can be modelled by a Gaussian distribution. We approximate the mean and variance of this distribution by the sample mean and sample variance found experimentally. We then need to take into account additional variance coming from the uncertainty in the number of runs retained after postselection. With $N$ trials in total, standard deviation $\sigma$ (after postselecting), and probability $p$ of postselection, it turns out[46] that the variance of the sample mean is ($\sigma^2 / (pN))(1 + (1 - p)/(pN) + O(1/(pN)^2))$ (see Supplementary Note 6 for a proof).

We now have error bars for the "raw" observable values produced after postselection, but before the other error mitigation techniques. As it is not straightforward to understand the effect of the TFLO procedure on errors analytically, we produce error bars for observables after TFLO using a Monte Carlo technique, where we assume that raw observables are distributed according to Gaussians with means and variances determined by the previous step. We then sample observables from these distributions 1000 times for each of the parameter settings used in TFLO (i.e. the FLO points and the VQE ground state point) and run the TFLO procedure to produce an energy estimate. The error bar we report is then the sample standard deviation of this estimate.

In the cases of quantities derived from expectations of multiple observables (i.e. spin and charge correlations) we make the simplifying assumption that the distribution of each of the observables combined to produce that quantity is independent to produce an overall error bar.

Error bars for the energies reported by BayesMGD are the internal estimates produced as described above. These show the level of certainty of the algorithm but may not correspond to a true error bar for the energy, if it were measured at the current parameters.

## Data availability

Data for these experiments are available at Ref. [47].

## Code availability

Code for these experiments is available at Ref. [47].

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

## Acknowledgements

This project has received funding from the European Research Council (ERC) under the European Union's Horizon 2020 research and innovation programme (grant agreement No. 817581) (A.M.) and from EPSRC grant EP/S516090/1 (J.L.B.). Google Cloud credits were provided by Google via the EPSRC Prosperity Partnership in Quantum Software for Modeling and Simulation (EP/S005021/1). We would like to thank members of the Phasecraft and Google Quantum AI teams for helpful suggestions, and in particular Toby Cubitt, Ryan Babbush, Yu Chen, Charles Neill and Pedram Roushan. We would also like to thank Benjamin Chiaro, Brooks Foxen, and Kevin Satzinger for their work on building and maintaining the Rainbow processor.

## Author contributions

A.M. and S.S. designed and implemented the quantum algorithms and error-mitigation procedures. J.L.B. designed, implemented and evaluated the classical optimisation algorithms. A.M., J.L.B. and S.S. ran the experiments. F.G. and R.S. proposed experiments and interpreted the results. E.O., T.E.O'B. and W.M. provided technical guidance and support for the quantum hardware. All authors contributed to drafting and editing the paper.

## Competing interests

A.M. is a cofounder of Phasecraft Ltd. A.M. and S.S. filed patent applications EP22153876.2, 17/586,260, GB2101242.2 on the TFLO method for error mitigation. All other authors declare no competing interests.
