## [Peer Review File · Nature Communications]

Observing ground-state properties of the Fermi-Hubbard model using a scalable algorithm on a quantum computerREVIEWER COMMENTS

Reviewer #1 (Remarks to the Author):

In this paper, the authors perform VQE simulations for the Fermi-Hubbard model in the 1×8 , 2×4 , and 1×4 lattices by resolving the occupation numbers, using up to 16 qubits on the 23-qubit Rainbow chip. The efficient Hamiltonian variational (EHV) ansatz developed by some of the authors, which is low depth and respects several symmetries of the Hamiltonian, allows the authors to achieve a significant improvement for mitigating errors for the largest system size ever performed on NISQ. The $U(1)$ symmetry corresponding to the particle-number conservation allows the authors to mitigate the qubit readout errors by postselection. Other symmetries allow for averaging the results over the symmetry sectors that are connected via the corresponding symmetry operations.

This work provides many useful techniques and insights for simulating ground-state properties of interacting fermions using NISQ devices, and will be an important step toward simulating quantum many-body systems on quantum computers. In principle, I am positive about recommending the paper for publication. However, before recommending the paper for publication, the authors should address the following points.

1. The authors mention that, in theory, the fidelity ~ 0.77 is achieved for the 1×8 lattice at half filling. I would suggest that, in the Supplementary Information (SI), the authors give the definition of the fidelity and show the 'Simulated' and 'Slater-determinant' fidelities vs N_{occ} for the 1×8 , 2×4 , and 1×4 lattices.
2. According to the results of the on-site spin correlation function $C^s(i, i) (= 1 - 2 \langle n_{i,\uparrow} n_{i,\downarrow} \rangle$ at half filling) shown in Fig.6, the VQE state overestimates the double occupancy as compared to the Ground state at half filling. On the other hand, the excess potential energy due to the larger double occupancy alone may not explain the total energy at half filling, and it is likely that the excess potential energy is compensated by the kinetic energy to some extent.

Now, it would be helpful for the reader to provide, possibly in the SI, both the kinetic and the potential energies as functions of N_{occ} for the 1×8 , 2×4 , and 1×4 lattices, to see how the relatively large error in energy near half filling appears. Showing such quantities, naturally evaluated in the VQE scheme for the Fermi-Hubbard model, would be helpful to further corroborate the important remark by the authors that the energies alone do not certify that the prepared state is a good approximation of the ground state.

3. SI Appendix D: N around Eq.(D3) is not defined (note that N is the number of trials in Sec. IV C, or of the i.i.d random variables in Appendix F). If L is the number of the lattice sites, then N here would be $2L$.

Then, I suppose that the particularly simple form $X^{\otimes N}$ of the particle-hole transformation operator \mathcal{P} is achieved thanks to the "snake" ordering (in 2D) on a bipartite lattice, where the sublattices A and B appear alternately along the snake line (otherwise \mathcal{P} may be given as a product of both X and Y operators). So I think it is worth reminding, in SI, of the choice of the Jordan-Wigner ordering the authors made.

4. In Fig.1, the onsite gate is depicted as "controlled-O" gate, but not in Fig.7. Depicting the same gate in the same manner would be preferable.
5. Giving the matrix representation and an explicit operator form ($\exp \left[i \frac{\pi}{8} (X_i X_j + Y_i Y_j) \right]$?) of the \sqrt{i} SWAP gate would be helpful.
6. SI Appendix E: Explicitly writing the expectation values and the variances of X_i and Y_i in terms of μ , σ , and p as equations would be helpful to follow the proof. In addition, X_0 in Eq. (F3) is not defined.
7. Sometimes the same symbol is used to represent the different quantities. For example, p represents the success probability of a Bernoulli trial (SI Appendix F) and the number of evaluation points in each optimization step (SI Appendix H). Please review the notation throughout the paper.

Reviewer #2 (Remarks to the Author):

This manuscript considers the Fermi-Hubbard model and demonstrates experimentally that current technology quantum computers can be useful in simulating small instances. The Fermi-Hubbard model has been suggested to be one of the simplest problems of significant practical relevance that could be solved with near-term quantum computers and indeed small-scale experimental demonstrations have already been presented.

The work goes beyond the prior art in several ways. First, a system of 16 qubits is simulated which is significantly larger than in prior works. Second, the work demonstrates successfully the applicability of some simple error mitigation techniques. Third, a good agreement between theory and experiments is demonstrated.

I think this work could be significantly improved by putting it more into context. In particular, I think the authors should reflect on how VQE compares with classical state-of-the-art approximations and they should also reflect on how scalable the VQE approach is; errors are mitigated significantly, but what is the associated increase in sampling costs and how scalable is it? I think these should ultimately give an idea to the reader where quantum computers are at present and how close/far we are from solving practical problems. If these comments are addressed I can recommend publication of this work in Nature Communications.

Detailed comments:

First, in Fig. 3 VQE optimisation results are presented. In Fig 3. (a) in the inset it looks as if the estimated energy were increasing during the optimisation process. Is this an artefact of errors/shot noise? Can the authors provide an intuitive explanation?

In Fig. 3 (b) the authors compare results of a number of error mitigation techniques, but I find there is a complete lack of a discussion as to how practical these techniques are. First, would it be possible to plot Fig. 3 (b) on a logarithmic scale – it is difficult spot any difference between the mitigation techniques. Second, is it possible to include the errors without mitigation? I think the authors should also discuss the following points. It seems in some instances some techniques even increase errors -- can this be explained? For symmetry verification one needs to discard samples – what was the associated cost/sampling overhead of the individual techniques?

In Fig. 4 results are presented based on which the authors conclude they show a good quantitative agreement. These plots confirm that the VQE results are generally better than a single Slater determinant approximation, however, I do not think they really speak to whether the VQE results are competitive in the following sense. I would expect a fair comparison should relate the VQE results to either state-of-the-art classical approximations or to some absolute scale. For example, in analogous quantum chemistry computations one could compare to coupled cluster or DFT techniques or against the absolute chemical precision. I think the authors should reflect on how competitive their results are when compared to classical computations.

Regarding scalability, I think it should also be discussed what was the sampling overhead associated with performing error mitigation. Would it be possible to reflect on how scalable is the approach in this sense, i.e., an estimate of the sampling overhead for larger systems, e.g., at 32 qubits or 64? Can the authors comment on the applicability of more advanced error mitigation techniques, such as exponential error suppression or virtual distillation – is it expected these could further reduce errors?

The authors introduce a new optimisation scheme that they term “BayesMGD” which fits a quadratic polynomial approximation to the VQE energy landscape and uses this information in the optimisation. This approach is compared to a few standard classical optimisation techniques. My main issue here is if the authors want to claim this paper introduces a new optimisation technique than they should compare it to the state-of-the-art from the VQE literature. Furthermore, Analytic Descent fits a classical approximation to the surface that is motivated by the true analytical form of the energy surface and is thus more “quantum aware” than a generic quadratic polynomial – and also provides an analytical approximation to the uncertainty of the surface. I think at least this connection should be

discussed. On the other hand, the main objective of this work seems to be the experimental demonstration and I feel it may just lose focus by claiming (and properly analysing) further technical results.

Observing ground-state properties of the Fermi-Hubbard model using a scalable algorithm on a quantum computer: Response to referees

We would like to thank the referees for their careful reading of the paper and helpful comments. Replies to their specific points are below.

Referee 1

1. The authors mention that, in theory, the fidelity ~ 0.77 is achieved for the 1×8 lattice at half-filling. I would suggest that in the Supplementary Information (SI), the authors give the definition of the fidelity and show the 'Simulated' and 'Slater-determinant' fidelities vs N_{occ} for the 1×8 , 2×4 , and 1×4 lattices

We have included these results in a new Figure 4 in Appendix E. For completeness, we include both the best possible fidelity of any Slater determinant, and the fidelity of the Slater determinant that achieves the best energy. In almost all cases, the simulated VQE fidelity outperforms the best Slater determinant fidelity.

2. According to the results of the on-site spin correlation function $\langle C^s(i,i) \rangle$ ($s=1,2$ and $\langle n_{i,\uparrow} n_{i,\downarrow} \rangle$) shown in Fig.6, the VQE state overestimates the double occupancy as compared to the ground state at half-filling. On the other hand, the excess potential energy due to the larger double occupancy alone may not explain the total energy at half-filling, and it is likely that the excess potential energy is compensated by the kinetic energy to some extent.

Now, it would be helpful for the reader to provide, possibly in the SI, both the kinetic and the potential energies as functions of N_{occ} for the 1×8 , 2×4 , and 1×4 lattices, to see how the relatively large error in energy near half-filling appears. Showing such quantities, naturally evaluated in the VQE scheme for the Fermi-Hubbard model, would be helpful to further corroborate the important remark by the authors that the energies alone do not certify that the prepared state is a good approximation of the ground state.

We have added these detailed energy plots to a new subsection of Appendix G, together with a discussion of the behaviour of these errors. We note that the overall energy error cannot be obtained just by summing the errors from the kinetic and potential energies, as our error mitigation methods act on the energy as a whole.

3. SI Appendix D: N around Eq.(D3) is not defined (note that N is the number of trials in Sec. IV C, or of the i.i.d random variables in Appendix F). If L is the number of the lattice sites, then N here would be $2L$.

We have added a definition of N .

Then, I suppose that the particularly simple form $X^{\otimes N}$ of the particle-hole transformation operator \mathcal{P} is achieved thanks to the "snake" ordering (in 2D) on a bipartite lattice, where the sublattices A and B appear alternately along the snake line

(otherwise $\langle \mathcal{P} \rangle$ may be given as a product of both X and Y operators). So I think it is worth reminding, in SI, of the choice of the Jordan-Wigner ordering the authors made.

This is correct - we have clarified this.

4. In Fig.1, the onsite gate is depicted as "controlled-O" gate, but not in Fig.7. Depicting the same gate in the same manner would be preferable.

We have changed Figure 7 to be the same as Figure 1.

5. Giving the matrix representation and an explicit operator form ($\exp\left[i\frac{\pi}{8}\left(X_i X_j + Y_i Y_j\right)\right]$) of the $\sqrt{\text{SWAP}}$ gate would be helpful.

We have added this to Figure 7.

6. SI Appendix E: Explicitly writing the expectation values and the variances of X_i and Y_i in terms of μ , σ , and p as equations would be helpful to follow the proof. In addition, X_0 in Eq. (F3) is not defined.

X_0 should have been X_1 throughout Eq. (F3). We have fixed that and some brackets, and added a line to Eq. (F3) to make the following paragraph more obvious.

7. Sometimes the same symbol is used to represent the different quantities. For example, p represents the success probability of a Bernoulli trial (SI Appendix F) and the number of evaluation points in each optimization step (SI Appendix H). Please review the notation throughout the paper.

We have replaced p , when used for the number of samples, with n_p throughout.

Referee 2

1. First, in Fig. 3 VQE optimisation results are presented. In Fig 3. (a) in the inset it looks as if the estimated energy were increasing during the optimisation process. Is this an artefact of errors/shot noise? Can the authors provide an intuitive explanation?

This is correct - we attribute this to fluctuations in performance of the device over time, given that the exact energy continues to improve. However, it is difficult to be definitive. We have highlighted this behaviour in the main text and added a few words about this. As a local, gradient-based optimiser which is constantly updating its parameters, our optimiser is immune to certain global fluctuations of the optimisation landscape, for example shifting by an overall additive or multiplicative constant.

2. In Fig. 3 (b) the authors compare results of a number of error mitigation techniques, but I find there is a complete lack of a discussion as to how practical these techniques are. First, would it be possible to plot Fig. 3 (b) on a logarithmic scale – it is difficult spot any difference between the mitigation techniques. Second, is it possible to include the errors without mitigation?

We agree that it is helpful to include errors without any mitigation. We have experimented with using a logarithmic scale, but feel that it is more confusing, because a) it necessitates switching to considering absolute errors, which loses information about whether an error-mitigation

technique overshoots or undershoots the true energy; b) error bars become more distracting and harder to interpret; c) several of the plots are close in absolute error. However, we agree that including the postselected and raw errors leads to the other techniques being too hard to distinguish. So we have updated the figure to plot the two “high-error” cases (postselected and raw) in a separate inset. We feel that this makes the plot more informative while still being easy to read.

I think the authors should also discuss the following points. It seems in some instances some techniques even increase errors -- can this be explained?

Yes, this is expected. An explanation for this is that, as our error-mitigation techniques are based on the use of additional data collected at other parameter values, and then used via, for example, averaging or linear interpolation, if the initial result happened to be accurate but the new data is less so, the mitigated value can be worse.

For symmetry verification one needs to discard samples – what was the associated cost/sampling overhead of the individual techniques?

We have added a mention of the sampling overhead of postselection in the main text, pointing to a longer discussion of the overhead of this and other techniques in Appendix D.

3. In Fig. 4 results are presented based on which the authors conclude they show a good quantitative agreement. These plots confirm that the VQE results are generally better than a single Slater determinant approximation, however, I do not think they really speak to whether the VQE results are competitive in the following sense. I would expect a fair comparison should relate the VQE results to either state-of-the-art classical approximations or to some absolute scale. For example, in analogous quantum chemistry computations one could compare to coupled cluster or DFT techniques or against the absolute chemical precision. I think the authors should reflect on how competitive their results are when compared to classical computations.

Although the focus of this work was on qualitative physical features that can be obtained from VQE, we have added a discussion about a quantitative comparison with state of the art methods in appendix I. Interestingly, these achieve a level of accuracy in terms of energy per site (estimated at ~ 0.03 for a system with 64 sites) which is comparable to the level of accuracy achieved by VQE, when compared with a perfect classical simulation. This suggests that while our current experiments can be simulated classically, maintaining this level of accuracy, while increasing the number of VQE layers and system size, will be sufficient to match or outperform classical methods.

4. Regarding scalability, I think it should also be discussed what was the sampling overhead associated with performing error mitigation. Would it be possible to reflect on how scalable is the approach in this sense, i.e., an estimate of the sampling overhead for larger systems, e.g., at 32 qubits or 64? Can the authors comment on the applicability of more advanced error mitigation techniques, such as exponential error suppression or virtual distillation – is it expected these could further reduce errors?

We added a new subsection about sampling overhead and scalability for the various error-mitigation techniques used at the end of Appendix D. We estimated the probability of

readout error for larger sizes to compute estimates of the cost of postselection by occupation number - even for a 128-qubit system, at half-filling we estimate a probability of retaining each run of ~7%. Finally, we added a comment about these other error-mitigation methods. These could be combined with the techniques we use, but experiments would be required to determine the level of improvement achieved.

5. The authors introduce a new optimisation scheme that they term “BayesMGD” which fits a quadratic polynomial approximation to the VQE energy landscape and uses this information in the optimisation. This approach is compared to a few standard classical optimisation techniques. My main issue here is if the authors want to claim this paper introduces a new optimisation technique than they should compare it to the state-of-the-art from the VQE literature. Furthermore, Analytic Descent fits a classical approximation to the surface that is motivated by the true analytical form of the energy surface and is thus more “quantum aware” than a generic quadratic polynomial – and also provides an analytical approximation to the uncertainty of the surface. I think at least this connection should be discussed. On the other hand, the main objective of this work seems to be the experimental demonstration and I feel it may just lose focus by claiming (and properly analysing) further technical results.

We believe that the key novelty of the BayesMGD algorithm is the use of Bayesian methods to update a surrogate model throughout the algorithm, rather than the particular choice of model (quadratic functions). The algorithm may be further improved by using a more sophisticated surrogate model that is informed by the analytical form of the cost function, and it is an interesting question for future research whether there could be an increase in performance by incorporating the “quantum aware” functions used in Analytic Descent into BayesMGD. We have added a discussion of this point to the end of Section 4D. We agree that the focus of this work is the experimental demonstration, as opposed to a comprehensive comparison of variational optimisers. We also remark that Analytic Descent assumes that the ansatz gates are generated by Pauli matrices, which is not the case here, so it is not obvious how to carry out such an experimental comparison.

REVIEWER COMMENTS

Reviewer #1 (Remarks to the Author):

In the revised version of the manuscript, the authors addressed this reviewer's comments and questions satisfactorily. I recommend the manuscript for publication in Nature Communications.

Reviewer #2 (Remarks to the Author):

I would like to thank the authors for carefully addressing all referee comments. Most of my reservations have been resolved, however, I still wonder about one point. To support the scalability of the presented experiments, the authors estimated error mitigation overheads (around 7% success rate at 128 qubits) based on a simple model that only takes into account readout errors. It suggests to me that the presented experiments are dominated by measurement errors which seems to be supported by the statement "many errors occur due to incorrect qubit readout, a significant source of error in superconducting qubit systems".

On the other hand, I would expect that if one scales up the system size then even at a constant number of ansatz layers (constant depth) the number of gate operations grows proportionally with the number of qubits. Depending on the depth, I would expect gate errors dominate measurement errors, e.g., in Google's quantum supremacy experiment the fidelity at 53 qubits was on the order of 0.1%. Can the authors roughly estimate the fidelity in the presented experiments and reflect on how this fidelity affects the efficiency of error mitigation, and how this would scale for larger problem sizes?

Observing ground-state properties of the Fermi-Hubbard model using a scalable algorithm on a quantum computer: Response to referees (2)

We would like to thank the referees for their further feedback on our paper. Our reply to the remaining comment of Referee 2 is below.

Referee 2

I would expect that if one scales up the system size then even at a constant number of ansatz layers (constant depth) the number of gate operations grows proportionally with the number of qubits. Depending on the depth, I would expect gate errors dominate measurement errors, e.g., in Google's quantum supremacy experiment the fidelity at 53 qubits was on the order of 0.1%. Can the authors roughly estimate the fidelity in the presented experiments and reflect on how this fidelity affects the efficiency of error mitigation, and how this would scale for larger problem sizes?

We agree that, as the system size increases, (2-qubit) gate errors will become the dominant type of errors in the quantum circuit. We have added a discussion of the effect of gate errors on p8 of the Supplementary Information, including quantitative estimates of fidelities in our experiments, details of quantum circuit complexities, and what one might expect to achieve for larger instances. It is challenging to provide rigorous statements about expected performance and overheads of error mitigation for larger systems - for example, our TFLO method should continue to be useful in a lower-fidelity regime, but determining the extent to which it can continue to reduce errors would require larger-scale experiments. We believe that, as demonstrated in our experiments, qualitatively accurate experimental results may still be achievable even with relatively low fidelities.

REVIEWERS' COMMENTS

Reviewer #2 (Remarks to the Author):

I would like to thank the authors for addressing all referee comments. I recommend publication in Nature Communications.